# Pilot study of a smartphone-based tinnitus therapy using structured counseling and sound therapy: A multiple-baseline design with ecological momentary assessment

Milena Engelke[1]*, Jorge Simões[1], Carsten Vogel[2], Stefan Schoisswohl[1], Martin Schecklmann[1], Stella Wölflick[1], Rüdiger Pryss[2], Thomas Probst[3], Berthold Langguth[1], Winfried Schlee[1,4]

1 Department of Psychiatry and Psychotherapy, University of Regensburg, Regensburg, Germany,
2 Institute of Clinical Epidemiology and Biometry, University of Würzburg, Würzburg, Germany,
3 Department for Psychotherapy and Biopsychosocial Health, Danube University Krems, Krems, Austria,
4 Institute for Information and Process Management, Eastern Switzerland University of Applied Sciences, St. Gallen, Switzerland

* milena.engelke@ukr.de

**Data Availability Statement:** The data that support the findings of this study are openly available on

## Abstract

Tinnitus affects a considerable part of the population and develops into a severe disorder in some sufferers. App-based interventions are able to provide low-threshold, cost-effective, and location-independent care for tinnitus patients. Therefore, we developed a smartphone app combining structured counseling with sound therapy and conducted a pilot study to evaluate treatment compliance and symptom improvement (trial registration: DRKS00030007). Outcome variables were Ecological Momentary Assessment (EMA) measured tinnitus distress and loudness and Tinnitus Handicap Inventory (THI) at baseline and final visit. A multiple-baseline design with a baseline phase (only EMA) followed by an intervention phase (EMA and intervention) was applied. 21 patients with chronic tinnitus ($\geq$ 6 months) were included. Overall compliance differed between modules (EMA usage: 79% of days, structured counseling: 72%, sound therapy: 32%). The THI score improved from baseline to final visit indicating a large effect (Cohens $d$ = 1.1). Tinnitus distress and loudness did not improve significantly from baseline phase to the end of intervention phase. However, 5 of 14 (36%) improved clinically meaningful in tinnitus distress ($\Delta$Distress $\geq$ 10) and 13 of 18 (72%) in THI score ($\Delta$THI $\geq$ 7). The positive relationship between tinnitus distress and loudness weakened over the course of the study. A trend but no level effect for tinnitus distress could be demonstrated by a mixed effect model. The improvement in THI was strongly associated with the improvement scores in EMA of tinnitus distress ($r$ = -0.75; 0.86). These results indicate that app-based structured counseling combined with sound therapy is feasible, has an impact on tinnitus symptoms and reduces distress for several patients. In addition, our data suggest that EMA could be used as a measurement tool to detect changes in tinnitus symptoms in clinical trials as has already been shown in other areas of mental health research.

GitHub (https://github.com/MilenaEn/UNITI-SCED-paper). DOI: 10.5281/zenodo.6958300.

**Funding:** This project has received funding from the European Union's Horizon 2020 Research and Innovation Programme, Grant Agreement Number 848261 (ME, JS, CV, SS, MS, SW, RP, BL, WS). TP didn't receive any funding. The funders had no role in study design, data collection and analysis, decision to publish, or preparation of the manuscript.

**Competing interests:** The authors have declared that no competing interests exist.

## Author summary

Tinnitus is the perception of a phantom sound which is generated in the brain. Many people are affected by tinnitus and some suffer severely. Management strategies can effectively reduce suffering of some patients. However, the access towards these treatments is limited due to long distances and lack of trained therapists. Therefore, there is increasing interest whether tinnitus management can be delivered via smartphone. In this study, we evaluated a smartphone application consisting of educational counseling and sound therapy with respect to treatment compliance and symptom improvement. We measured tinnitus symptoms at the beginning of baseline and at the end of intervention using clinical questionnaires, but also during the course of treatment using daily questions which were integrated in the mobile application. The app was used regularly by the patients (sound therapy: 32% of days, structured counseling: 72% of days; tinnitus assessment: 79% of days). Tinnitus handicap improved clinically meaningful for 13 of 18 patients, tinnitus distress for 5 of 14 and tinnitus loudness for 2 of 14. The correlation between tinnitus distress and loudness weakened during the study course. Thus, this study indicates that app-based tinnitus management has an impact on tinnitus symptoms and reduces suffering for several patients.

## Introduction

Experiencing subjective tinnitus implies an involuntary perception of a phantom sound that is generated by abnormal neural activity in the auditory system [1,2]. Around 14–15% of the population in European countries are affected by tinnitus, while 1–2% are severely impaired by tinnitus [3,4]. Tinnitus is a heterogeneous condition, not only in terms of level of impairment, but also regarding its perceptual characteristics, etiological factors, pathological markers, clinical comorbidities, and its treatment response [2,5,6]. Psychological, auditory, and neurological treatment approaches exist that aim towards reduction of tinnitus symptoms and improvement of life quality, but up to now, there is no universal cure for tinnitus available for the clinical routine [1,7].

Patients who seek health care usually suffer from severe tinnitus which is accompanied by emotional distress, cognitive dysfunction and increased arousal ultimately leading to functional disability and reduced life quality [2,3]. Among others, patients complain sleep disturbances, problems in concentration, attention, communication, and memory and some are severely affected by mental disorders [8,9]. Tinnitus severity is mainly influenced by psychological factors. High life stress, poor coping resources, co-morbid mental diseases and personality dimensions shape negative emotional reactions, selective attention, and cognitive appraisal towards the tinnitus. Involving numerous feedback loops, tinnitus distress is likely to persist which in turn prevents habituation to the new stimulus [10–12]. Tinnitus management strategies aim towards stimulation of the habituation process by reducing stress, anxiety, negative emotional responses, and dysfunctional cognitions on one hand, and by enhancing coping resources on the other hand [13,14].

Despite scientific efforts, the health care situation for tinnitus patients is deficient, since the access to available treatment options for tinnitus is limited [15]. In addition to the lack of trained therapists, environmental factors such as the COVID-19 outbreak in 2020 can further reduce access to face-to-face treatments [16–18]. Moreover, tinnitus has a substantial economic burden in terms of health care and indirect costs with increasing trend [3,19,20].

In response to the deficient health care situation, smartphone-based therapies are capable to provide low-threshold (easily accessible), cost-effective and location-independent healthcare and can potentially increase patients treatment compliance [15,21,22]. They are particularly beneficial for the management of chronic diseases as they strengthen patient empowerment and are able to induce behavior modification and lifestyle changes [21,22]. Furthermore, smartphone-based therapies can be used to provide information about tinnitus to a large number of affected people, as well as to track the course of tinnitus symptoms in a longitudinal, ecologically valid, and momentary fashion [15,23,24]. Currently, there are many available smartphone applications for tinnitus, but evidence for their efficacy is limited [15,25]. Preliminary findings indicate that app-based treatments have the potential to improve tinnitus symptoms [25–30]. Most evaluated apps use Cognitive Behavioural Therapy (CBT) approaches and auditory treatments; to the best of our knowledge, there is currently no evaluation of an app-based structured counseling combined with sound therapy.

Therefore, we developed a tinnitus management app that provides structured counseling supported by sound therapy. Structured counseling has been shown to improve tinnitus distress and quality of life in face-to-face settings or through written manuals [31,32] and may have better effects when combined with sound therapy [32–34]. Additionally, the app contains an Ecological Momentary Assessment (EMA) module that tracks changes in tinnitus symptoms on a daily basis. The feasibility of EMA to measure the course of tinnitus has already been investigated in several studies [23,24,35] and has also been used to evaluate tinnitus treatments [36].

Hence, we conducted a pilot-study to evaluate the app-based treatment. The objective was to investigate treatment adherence and improvement of tinnitus symptoms individually and on a group level. A trend effect (improvement over time) for daily measured tinnitus distress was hypothesized. Furthermore, we analyzed the association of the two measurement methods, namely EMA and clinical questionnaires. To account for tinnitus heterogeneity and individual treatment effects, a multiple-baseline across groups design was chosen. Multiple-baseline designs belong to the group of single-case experimental designs (SCED) where the outcome variable is measured longitudinally before (baseline phase) and during the intervention (intervention phase) in single cases. Hence, with the baseline phase, each case has its own control and is considered as replication [37]. As SCEDs can be performed with small sample sizes, they are especially recommended for piloting novel interventions [38]. In multiple-baseline designs, the introduction of the intervention phase is staggered in time to separate treatment effects from random time effects and therefore enhance internal validity [37].

## Materials and methods

### Study design

A randomized multiple-baseline design across groups was used. Participants first underwent a baseline phase where tinnitus symptoms were measured daily using app-based EMA, followed by an intervention phase where the intervention (structured counseling and sound therapy) was additionally introduced. Participants were allocated to 4 groups with different baseline lengths to realize staggered introduction of the intervention and to fulfil the recommendations of at least 3 replications [38]. The baseline phase had a minimum duration of 7 days (i.e. 7 EMA data points) to exceed the recommended standard of 5 data points per phase [39]. Furthermore, the groups entered the intervention phase with a distance of 3–5 days to each other to keep baseline phase as short as possible. As randomization is recommended for SCEDs to minimize threats to internal validity, the Koehler-Levin randomization procedure was used to randomize the intervention start [40,41]. Therefore, 2 potential starting points for each group

were defined considering the 3–5 days distance to the other groups. First, the participants were randomly assigned to the 4 groups. Then, the order of the groups was randomly assigned and finally, one out of two potential starting points was randomly selected for each group. The intervention phase had the same duration of 12 weeks for all participants. At the beginning of the baseline phase and at final visit (after intervention phase), participants completed online versions of clinical questionnaires about tinnitus and other health-related aspects using an electronic database. The study was designed in line with SCED recommendations [39].

The study is approved under the ethic vote 20–1936_2–101 „UNification of treatments and Interventions for TInnitus patients–Randomized Clinical Trial (UNITI-RCT)"(Amendment No. 2) by the ethics committee of the University of Regensburg, Germany [42]. All persons were informed about the objectives and methodologies of the present study, and gave written informed consent prior to study start. Participants received neither compensation nor incentives for adherence to the study protocol. The study was conducted according to the guidelines in the Declaration of Helsinki. It was retrospectively registered in the German Clinical Trials Register (DRKS-ID: DRKS00030007) because of lack of awareness that pilot trials need to be registered. The CONSORT checklist (S1 Checklist) and the study protocol (S1 Protocol) are part of the supporting information.

## Participants

Inclusion criteria were: 1) adult (age $\geq$ 18 years), 2) chronic tinnitus (duration $\geq$ 6 months), and 3) THI-score $\geq$ 18 (indicating at least a mild tinnitus handicap) [43]. We included patients with any tinnitus type (e.g., subjective, objective, somatosensory) regardless of comorbidities (e.g., hearing loss, hyperacusis). Exclusion criteria were: 1) tinnitus-related therapy within the last 3 months and 2) similar therapy with long term effects e.g., noiser. Further, potential participants had to own a smartphone with the operating systems iOS 12.0 (or newer) or Android 5.0 (or newer) in order to use the mobile application as part of the study.

## Procedures

The whole study was conducted online during the COVID-19 pandemic (May 2021 –October 2021). Patient communication was realized by mail. By giving their consent, participants were included into the study and randomized to one of the four groups. Also, they were provided online via a database with the clinical questionnaires before the baseline phase. With the start of the baseline phase on the same day for all participants, information about app installation and login data were provided. During baseline phase, participants were instructed to answer the diary (EMA module) every evening. With the start of the intervention phase, participants received instructions and new login data that enabled access to the treatment modules. All participants immediately started the intervention phase on the defined day. Apart from answering the diary, participants were instructed to complete daily sessions of structured counseling and sound therapy during a 12-week intervention phase. At halftime (after 6 weeks of intervention), a reminder to fulfill the study protocol (daily usage of intervention and diary) was sent out. The intervention phase ended after 12 weeks with the final visit assessment.

## App-based intervention

Both EMA recordings and the intervention modules (structured counseling and sound therapy) were provided via the smartphone-based application UNITI running on the operating systems iOS 12.0 or newer and Android 5.0 or newer. It is available in the Apple App Store and the Google Play Store. App architecture is described elsewhere [44].

## Structured counseling

The structured counseling module *TinEdu* consists of 12 chapters, each chapter includes 7 sections and a quiz at the end. The sections contain informative texts, related links, illustrations, and audio files imparting knowledge about tinnitus ranging from basics about hearing and tinnitus to more advanced topics such as therapeutic options and current tinnitus research. In addition, practical tips are provided on how to deal with tinnitus on a daily basis. The last section of each chapter gives a short summary of the provided content. After that, participants can test themselves in a subsequent quiz. The quiz consists of single- and multiple-choice questions related to the respective chapter. Participants were provided with feedback on their test performance and were advised to repeat the relevant sections in case of a wrong answer. The Feedback section is a separate module of the UNITI app that functionally belongs to *TinEdu* and saves the feedback of the quizzes. As the intervention is scheduled for 12 weeks, one section should be completed each day so that a chapter is completed every week. To ensure the processing order of *TinEdu*, each section is unlocked only when the previous one has been completed.

## Sound therapy

The sound therapy *Shades of Noise* (*SoN*) forms the second treatment module of the UNITI application. It provides a sound library of 64 sounds from which 29 are naturalistic and 35 are digitally produced sounds. In the first category, sounds as beach waves or waterfall noise are presented. The second category consists of pure tones and broad band noises of different frequencies with optional filters and modulation rates representing the different sound therapeutic approaches (e.g., masking, residual inhibition). At the first use, subjects could optionally set their tinnitus frequency, if known, so that some sounds could be adjusted to the pitch. To increase the benefit of the therapy, it was recommended to listen to the sounds with headphones to achieve a more even distribution to both ears. When participants started a session, they had to indicate their current tinnitus loudness and distress on a visual analogue scale (VAS). After that, they could select any sound from the library to listen to. Different play times could be set, between 1 minute and infinite play time. Also, sounds could be marked as favourites, making them appear at the top of the sound list for the upcoming sessions. Furthermore, each sound could optionally be rated on how much the tinnitus loudness changed after listening to it. Participants were encouraged to use *SoN* in a self-guided manner with a recommendation of at least 15 minutes per day.

## Measures

Outcomes were measured with clinical questionnaires at the beginning of the baseline phase and at the final visit assessment after the end of the intervention provided via an electronic database, as well as with active EMA data using the diary via the smartphone-application. Data protection and data transfer was handled according to the highest security standards and has been approved by the Data Protection Officer (DPO) of the University of Regensburg, Germany.

Clinical questionnaires included the THI [43], the German Hyperacusis Questionnaire (GUEF) [45], the Patient Health Questionnaire for Depression (PHQ-9) [46], and the European School of Interdisciplinary Tinnitus Research Screening Questionnaire (ESIT-SQ) [47]. The THI consists of 25 items with 3 answer options addressing the impact of tinnitus on daily life. Higher THI score indicates higher tinnitus handicap [43]. The remaining questionnaires provide information on demographics, tinnitus characteristics, co-morbid hyperacusis and mental health. The questionnaire data were stored anonymized in an electronic database (EU

Tinnitus Database). The database is hosted by database experts from the University of Regensburg at the Bezirksklinikum Regensburg, where the server is located.

The EMA module (tinnitus diary) consisted of a collection of 10 VAS questions regarding current tinnitus perception and thoughts, neck and jaw tension, as well as psychological well-being. In an additional empty field, subjects were allowed to make notes on special occurrences. The EMA module was designed as an end-of-day diary to track changes in tinnitus symptoms evoked by the intervention, therefore, participants received daily notifications at 19:30. The authors are aware that the notifications might influence the time of app usage. For this study, results of two questions about tinnitus perception have been included in the analysis (see S1 Table). These questions assess the current tinnitus distress (*How burdensome do you find your tinnitus at the moment*?) and the current tinnitus loudness (*How loud is your tinnitus at the moment*?) on a continuous rating scale (0–100; *not burdensome/inaudible–very burdensome/very loud*). The app data are stored in a relational database using Maria DB 11, which runs in a Linux environment (Debian Buster) and on a LAMP technology stack. No personal information is stored there.

## Statistical analysis

All participants who filled in the 'begin of baseline' and 'final visit' questionnaires and used the app at least once during baseline and intervention phase were included in the data analysis. Descriptive statistics for demographics, tinnitus characteristics (retrieved from the clinical questionnaires) and app use were calculated to characterize the sample. A paired t-test was used to determine the sample difference in THI score from begin of baseline to final visit. The main analysis focus was on the EMA data provided by the app diary. Therefore, data were additionally prepared and missing values were imputed using the aregImpute function of the Hmisc R package with its default settings [48]. To support visualization of tinnitus symptom courses, individual means and standard deviation were calculated for baseline phase and end of intervention phase and statistically contrasted on a group level using a paired t-test. The change in relation between tinnitus distress and loudness was described with Pearson's product-moment correlation for each week. A mixed effect model was computed to predict tinnitus distress including demographics (age, gender), psychological burden (THI, PHQ-9) and intervention variables (trend, level) as fixed effects and participants as random intercepts. To compare the different measurement methods, Pearson's product-moment correlation between improvement in THI and improvement in tinnitus distress was calculated. Data preparation and analysis was conducted in R (version 4.1.2).

## Results

### Participants

A minimum of 20 participants were defined for inclusion to allocate at least 5 participants to each group. 40 potentially eligible patients were preselected from a patient pool based on inclusion criteria and contacted per mail (see Fig 1 for patient flow). 21 declared their consent, were included into the study and randomized to one of the 4 groups. The Koehler-Levin method was applied [40,41]. Group 1 was assigned to a baseline length of 11 days, group 2 of 19 days, group 3 of 16 days and group 4 of 8 days. Double-checking of the eligibility at begin of the baseline phase confirmed that all 21 participants fulfilled the inclusion criteria. For data analysis, 3 cases (3/21, 14%) had to be excluded, because of missing final visit questionnaires (case 02, case 09) and missing app data from the intervention phase (case 01). Hence, 18 patients are considered for data analysis.

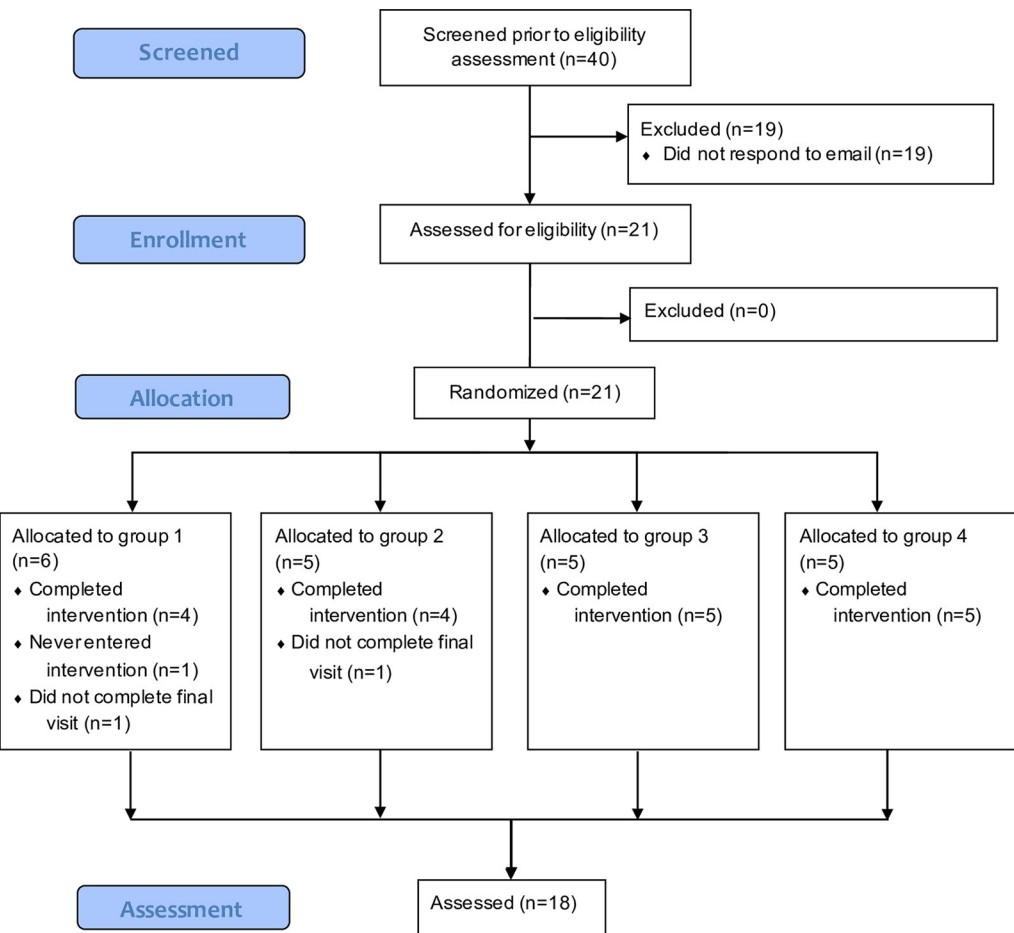

**Fig 1. CONSORT Flow Diagram.** Flow diagram of patient recruitment and dropouts. Study design: Multiple-baseline design across groups. The four groups introduced the treatment staggered in time to separate treatment effects from potential time effects and therefore enhance internal validity. Baseline length for each group was randomly selected following the Koehler-Levin procedure and patients were randomly allocated to groups. Intervention phase was the same length for all groups. Before baseline and after intervention phase, clinical questionnaires were applied.

## Demographics, tinnitus characteristics and app use

Table 1 summarizes demographics, tinnitus characteristics and app use. The sample was balanced in terms of gender, age and education level. PHQ-9 score at baseline indicated mild depressive symptoms on average [49]. The sample was moderately handicapped (THI) by their tinnitus, slightly to moderately impaired by hyperacusis (GUEF) and has had tinnitus for an average of 183 months [45,50]. 11 patients had seen a physician in the past year due to their tinnitus complaints. Patients had good overall compliance using the app. On average, the diary was used on 75.9 days (about 79% of days), structured counseling was used on 60.7 days (72% of days), and sound therapy on 26.6 days (32% of days). A daily usage was recommended (the intervention phase lasted for 84 days and the baseline phase for 8–19 days, depending on group affiliation).

## Change in THI

Change of THI score from the begin of baseline phase to final visit is depicted in Table 2 and visualized in Fig 2. The mean THI was improved by 11.8 points from begin of baseline to final

**Table 1. Participant demographics, tinnitus characteristics and app use.**

| *Participant demographics* | |
|---|---|
| Participants | *N* = 18 |
| Female, n (%) | 8 (44.4%) |
| Age in years, mean (range) | 49.8 (24–68) |
| Education, n | 2 (elementary school), 5 (secondary school), 3 (A-levels), 8 (university) |
| PHQ-9, mean (SD) | 5.7 (3.3) |
| *Tinnitus characteristics* | |
| THI, mean (SD) | 38.4 (15.4) |
| Tinnitus duration in months, mean (range) | 183 (20–350) |
| GUEF Hyperacusis, mean (SD) | 10.3 (6.7) |
| Vertigo, n | 4 (never), 6 (less than once a year), 8 (more than once a year) |
| Number of perceived sounds, n | 12 (1 sound), 6 (more than 1 sounds) |
| Tinnitus onset, n | 6 (slowly increasing), 8 (sudden), 4 (don't know) |
| Loudness changes, n | 5 (never), 11 (sometimes), 2 (always) |
| Tinnitus quality, n | 10 (tone), 5 (noise), 3 (other) |
| Tinnitus pitch, n | 12 (high), 5 (medium), 1 (deep) |
| Tinnitus lateralization, n | 6 (right), 3 (left), 8 (same for both ears), 1 (inside the head) |
| Tinnitus healthcare visits during the past year, n | 7 (0 visits), 7 (1 visit), 4 (2–4 visits) |
| *App use (in sessions)* | |
| Diary, mean (SD) | 75.9 (25.6) |
| TinEdu, mean (SD) | 60.7 (29.5) |
| Shades of Noise, mean (SD) | 26.6 (33.3) |

Description of the sample at baseline. Tinnitus characteristics were assessed with the European School of Interdisciplinary Tinnitus Research Screening Questionnaire (ESIT-SQ). PHQ-9: Patient Health Questionnaire for Depression; THI: Tinnitus Handicap Inventory; GUEF: German Hyperacusis Questionnaire. App use: Mean and standard deviation of the number of sessions.

visit indicating a large effect (Cohens *d* = 1.1). 13 of 18 cases (72%) improved clinically meaningful on the THI score ($\Delta$THI $\geq$ 7). A THI improvement of 7 points has been empirically determined as the minimal clinically relevant improvement [51].

## Change in EMA of tinnitus symptoms

EMA data from the tinnitus diary were prepared for visualization and analysis creating a structure of one data point per case per day. In case of multiple entries per day, the first one was kept (*n* = 79 duplicates removed). Cases were visually inspected regarding randomness of missing values, 4 cases had to be excluded because of data not missing at random (cases 03, 06, 08, 21). For the remaining 14 cases, missing data points were imputed with multiple imputation using bootstrapping to simulate the process of drawing predicted values from a full Bayesian predictive distribution (*n* = 1.510 imputed data points). The EMA subset (*N* = 14) is comparable with the full sample (*N* = 18) regarding participant demographics and tinnitus characteristics (see S2 Table).

## Visualization of tinnitus symptom courses

Fig 3 displays the course of daily measured tinnitus distress and loudness for each case over the study duration. The grey area represents the baseline phase (only EMA), the white area

**Table 2. Improvement of THI from begin of baseline phase to final visit.**

| Begin of baseline | | Final visit | | Paired t-test | | |
|---|---|---|---|---|---|---|
| **Mean** | **SD** | **Mean** | **SD** | **t(17)** | **p** | **Cohen's d [95% CI]** |
| 38.4 | 15.4 | 26.6 | 18.4 | 4.55 | < .001 | 1.1 [0.49, 1.69] |

Mean and standard deviation of the Tinnitus Handicap Inventory (THI) score at begin of baseline phase and final visit. Paired t-test reveals a statistically significant improvement from baseline to final visit with a strong effect size. N = 18.

represents the intervention phase (EMA, structured counseling, sound therapy). The heterogeneity between cases and the fluctuation of symptom courses hamper identification of symptom improvement from visualization, therefore, descriptive summaries and statistical analysis are performed in the next step.

## Improvement from baseline phase to the end of intervention phase

Next, we were interested if tinnitus symptoms improved from baseline phase to the end of the intervention phase. The end of the intervention phase is defined by the last days of the intervention phase using the same time window than the baseline phase (Group 1: 11 days, group 2: 19 days, group 3: 16 days, group 4: 8 days). Mean and standard deviation were calculated for the baseline phase and the end of the intervention phase. Individual values are depicted in Table 3 for tinnitus distress and Table 4 for tinnitus loudness. On a group level, no significant differences between baseline phase and end of intervention phase have been found neither for tinnitus distress (mean of the difference = 4.5; $t(13) = 1.08$, $p = 0.30$, 95% CI [-4.5;13.6]), nor for tinnitus loudness (mean of the difference = -2.2; $t(13) = -0.79$, $p = 0.45$, 95% CI [-8,1;3.8]). On an individual level, 5 of 14 cases (36%) improved clinically meaningful on distress

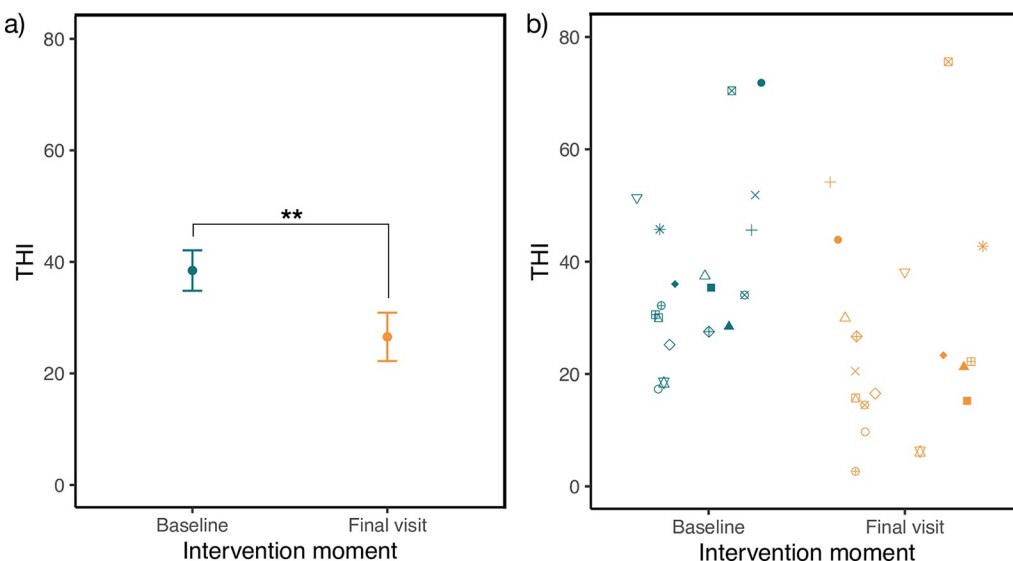

**Fig 2. Improvement of THI from begin of baseline phase to final visit.** a) Significant improvement of Tinnitus Handicap Inventory (THI) score from the begin of baseline phase to final visit of the sample. Higher scores indicate higher handicap. Dots represent means, error bars are standard errors; ** $p < .001$; N = 18. b) Individual THI values at baseline and final visit. The same shape represents the same individual.

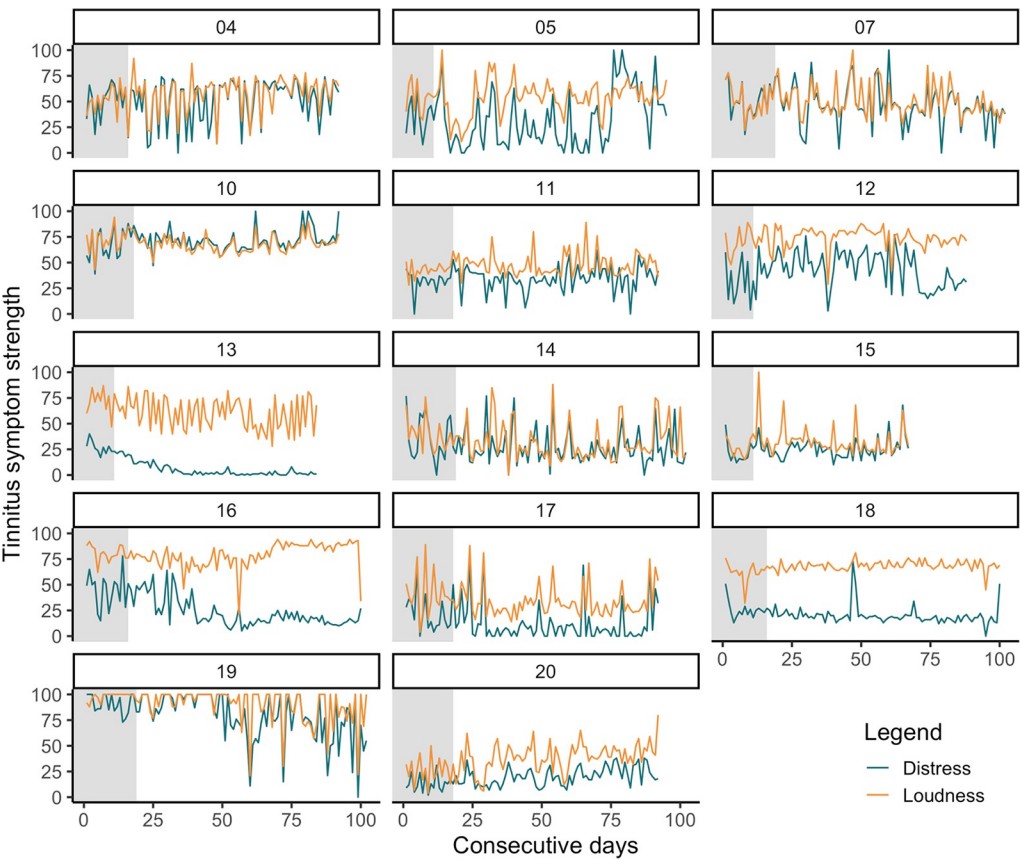

**Fig 3. Course of tinnitus distress and loudness for each case (EMA).** Individual courses of tinnitus symptoms that were assessed daily with Ecological Momentary Assessment over the study period. The grey area represents the baseline phase (EMA), the white area represents the intervention phase (EMA, structured counseling, sound therapy). Distress: How burdensome do you find your tinnitus at the moment? (0 not burdensome– 100 very burdensome); Loudness: How loud is your tinnitus at the moment? (0 inaudible– 100 very loud). $N$ = 14.

($\Delta$Distress $\geq$ 10). Those patients improved clinically meaningful on the THI score as well ($\Delta$THI $\geq$ 7, see Table 3). Regarding tinnitus loudness, only 2 of 14 cases (14%) improved clinically meaningful from baseline phase to the end of intervention phase ($\Delta$Loudness $\geq$ 10, see Table 4). An improvement of 10 points has been empirically determined as the minimal clinically relevant improvement for tinnitus distress and loudness measured on a 100 point VAS [52].

## Change in relationship between tinnitus distress and loudness

Visualization of the individual symptom courses have led to the observation that the courses of tinnitus distress and loudness seem to diverge for some individuals (Case 13 & 16, see Fig 3). To quantify that observation, we calculated means for tinnitus distress and loudness for the baseline phase and for each of the 12 weeks of the intervention phase and correlated the weekly distress and loudness means with each other (see Fig 4). All correlations are positive and reach statistical significance ($p$ < .001). The strongest correlation was measured in week 2 of the intervention phase ($r$ = 0.72) and the weakest in week 11 ($r$ = 0.35). On a descriptive level, the correlation coefficients between tinnitus distress and loudness tended to decline over the course of the study.

**Table 3. Improvement in tinnitus distress from baseline to end of intervention phase.**

| Cases | Baseline phase | | End of intervention phase | | ΔDistress (EMA) | ΔTHI |
|---|---|---|---|---|---|---|
| | Mean | SD | Mean | SD | | |
| 04 | 52.0 | 15.5 | 53.8 | 16.0 | -1.8 | 8 |
| 05 | 35.0 | 17.5 | 48.1 | 22.0 | -13.1 | -8 |
| 07 | 50.6 | 15.4 | 38.5 | 13.3 | 12.1 | 10 |
| 10 | 63.3 | 16.9 | 74.1 | 10.8 | -10.8 | -6 |
| 11 | 31.1 | 14.6 | 42.9 | 9.4 | -11.8 | 4 |
| 12 | 32.3 | 21.0 | 30.6 | 6.5 | 1.7 | 2 |
| 13 | 26.8 | 6.5 | 2.2 | 2.2 | 24.6 | 30 |
| 14 | 36.1 | 21.5 | 25.4 | 18.1 | 10.7 | 12 |
| 15 | 21.3 | 11.2 | 31.7 | 15.4 | -10.4 | 8 |
| 16 | 44.1 | 17.5 | 14.3 | 4.3 | 29.8 | 20 |
| 17 | 28.4 | 20.8 | 21.1 | 25.3 | 7.3 | 14 |
| 18 | 25.4 | 8.8 | 18.2 | 10.1 | 7.2 | 20 |
| 19 | 91.2 | 9.4 | 60.7 | 27.9 | 30.5 | 28 |
| 20 | 12.7 | 8.5 | 25.3 | 10.3 | -12.6 | 6 |

Individual distribution of tinnitus distress (EMA) during baseline phase and end of intervention phase (last days according to baseline length). Higher values indicate higher distress. ΔDistress: Baseline phase–End of intervention phase; ΔTHI: THI begin baseline–THI final visit.

## Trend and level effect of tinnitus distress

A mixed effect model was only conducted for tinnitus distress, as prior findings indicate no effect for tinnitus loudness. In Table 5, the results are summarized. To evaluate the treatment effect for tinnitus distress, a trend and level effect was modeled [53]. The trend effect models

**Table 4. Improvement in tinnitus loudness from baseline to end of intervention phase.**

| Cases | Baseline phase | | Intervention phase | | ΔLoudness (EMA) |
|---|---|---|---|---|---|
| | Mean | SD | Mean | SD | |
| 04 | 52.3 | 12.1 | 62.6 | 10.8 | -10.3 |
| 05 | 53.2 | 15.5 | 56.5 | 7.9 | -3.3 |
| 07 | 50.8 | 15.3 | 42.3 | 8.4 | 8.5 |
| 10 | 68.9 | 14.5 | 69.8 | 3.9 | -0.9 |
| 11 | 41.6 | 9.3 | 51.0 | 8.0 | -9.4 |
| 12 | 70.2 | 13.7 | 71.5 | 3.6 | -1.3 |
| 13 | 71.2 | 12.1 | 58.6 | 19.9 | 12.6 |
| 14 | 39.5 | 15.2 | 35.7 | 19.5 | 3.8 |
| 15 | 25.8 | 8.0 | 31.5 | 12.7 | -5.7 |
| 16 | 80.9 | 7.8 | 85.6 | 14.1 | -4.7 |
| 17 | 37.9 | 23.0 | 41.9 | 20.5 | -4.0 |
| 18 | 61.9 | 9.7 | 68.4 | 7.1 | -6.5 |
| 19 | 97.9 | 3.8 | 81.9 | 22.1 | 16.0 |
| 20 | 25.9 | 12.1 | 51.0 | 14.3 | -25.1 |

Individual distribution of tinnitus loudness (EMA) during baseline phase and end of intervention phase (last days according to baseline length). Higher values indicate higher loudness. ΔLoudness: Baseline phase–End of intervention phase.

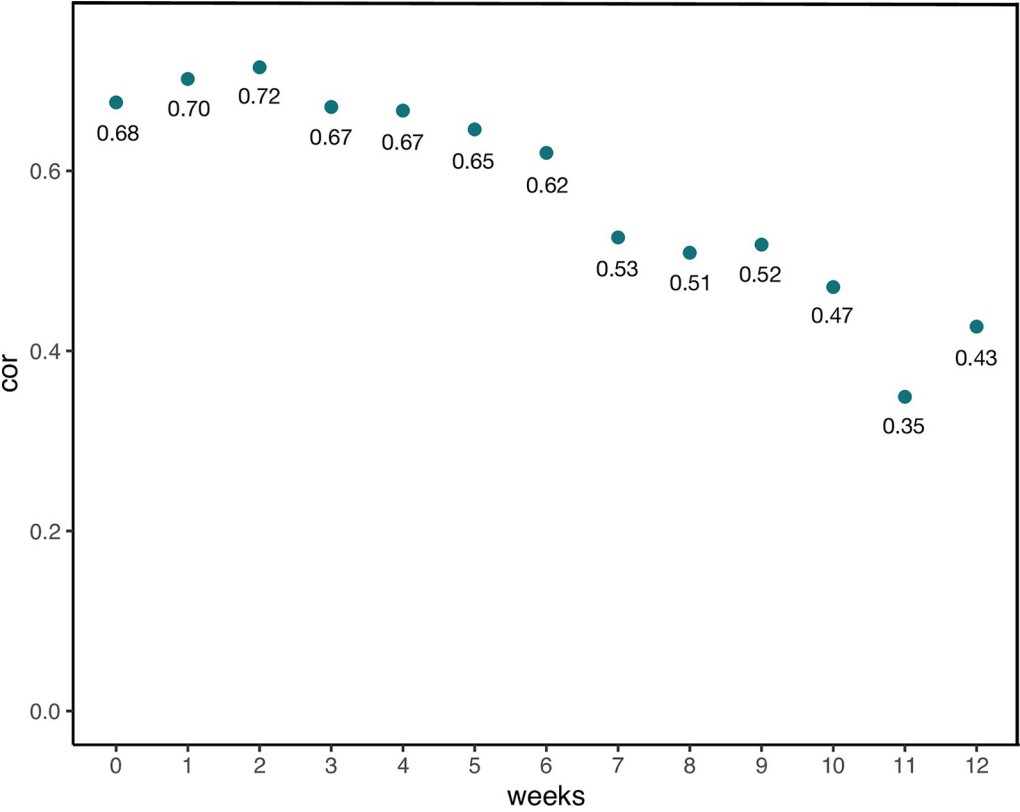

**Fig 4. Change of correlation between distress and loudness over the study period (EMA).** Pearson's product-moment correlations between tinnitus distress and loudness during baseline phase (week 0) and intervention phase (week 1–12). All correlations are statistically significant ($p < .001$). $N = 14$.

the change in tinnitus distress over time (in days), while the level effect models the change in distress between the phases (from baseline to intervention phase). Hence, trend and level were included as fixed effects. Besides from the intervention variables, demographics (age, gender) and psychological burden variables (THI, PHQ-9) were additionally included as fixed effects to enhance tinnitus distress prediction. Participants were included as random intercepts. The outcome variable was daily measured tinnitus distress.

The model explained 62% of the variance of daily tinnitus distress, mainly explained by the fixed effects (52% explained variance). On average, tinnitus distress was significantly reduced by 0.06 points per day (within the whole study period; trend effect) and descriptively reduced by 2.38 points from baseline to intervention phase (level effect). Moreover, cases with higher daily distress tended to be male with higher baseline THI and PHQ-9 scores.

To test the model for overfitting, the train/test split method was applied. The mixed effect model was fitted to the training data (80% of the data set) and tested on the test data. The explained variance of the test data was $R^2 = 0.67$ (conditional).

## Comparison of improvement in THI and tinnitus distress

As the THI score is a well-accepted clinical validation tool for tinnitus treatments, we were furthermore interested if improvement in EMA of tinnitus distress was associated with THI improvement and which EMA improvement score represents THI improvement best. Therefore, we defined two EMA improvement scores. The first score is the Distress trend (slope of tinnitus

**Table 5. Mixed effect model.**

| Fixed effects | Coefficient | Se | 95% CI | p |
|---|---|---|---|---|
| Intercept | -24.64 | 16.48 | -53.43–4.20 | 0.17 |
| Trend | -0.06 | 0.02 | -0.10 – -0.02 | < 0.01 |
| Level (int.) | -2.38 | 1.70 | -5.78–0.90 | 0.16 |
| Age | 0.38 | 0.27 | -0.09–0.85 | 0.18 |
| Gender (m) | 17.25 | 5.13 | 8.29–26.24 | < 0.01 |
| THI | 0.72 | 0.19 | 0.38–1.06 | < 0.01 |
| PHQ-9 | 1.52 | 0.56 | 0.45–2.53 | < 0.01 |
| **Model fit** | **$R^2$ con** | **$R^2$ marg** | | |
| | 0.623 | 0.515 | | |

Prediction of tinnitus distress using intervention variables (trend as time in days, level as difference from intervention to baseline phase), demographics (gender, age) and psychological burden (baseline THI, PHQ-9) as fixed effects and subjects as random effects. Model equation: Tinnitus distress ~ trend + level + age + gender + THI + PHQ-9 + (1|case). Dummy coding: Level (0 baseline phase, 1 intervention phase), Gender (0 female, 1 male). $N = 14$.

distress during intervention phase) and the second score is the ΔDistress used before (see Table 3, baseline phase–end of intervention phase). We compared the association of both EMA improvement scores with ΔTHI (THI begin baseline–THI final visit), the results are depicted in Fig 5. ΔTHI is strongly associated with both EMA improvement scores (Distress trend and ΔDistress), with a descriptively stronger association with ΔDistress (see Fig 5B; $r = 0.86$, $p < 0.001$).

## Discussion

We conducted a pilot study to investigate the feasibility and efficacy of an app-based tinnitus intervention combining structured counseling with acoustic therapy using a multiple-baseline across groups design with an initial sample of 21 participants. Outcome variables were EMA of

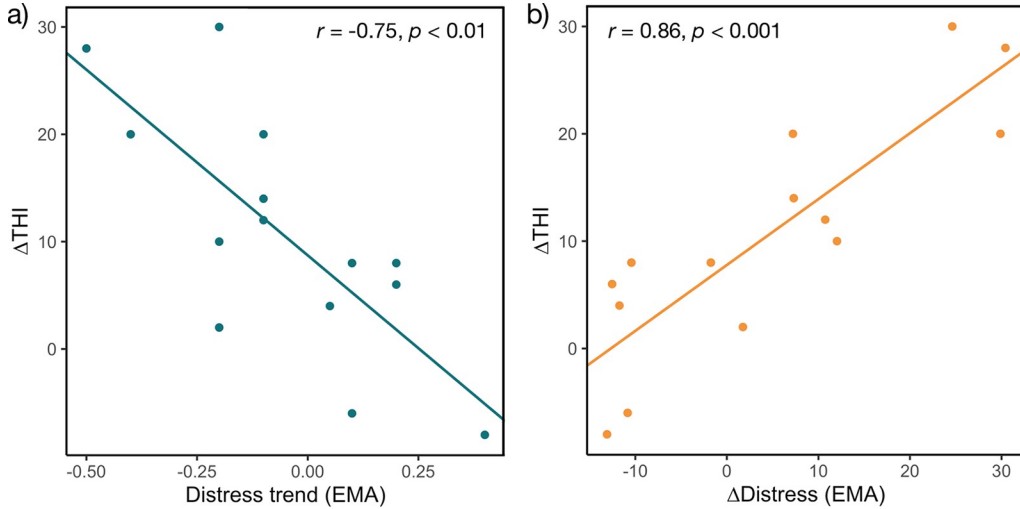

**Fig 5. Correlation of ΔTHI with improvement in EMA tinnitus distress.** Pearson's product-moment correlation of ΔTHI with two EMA improvement scores: Distress trend (a) and ΔDistress (b). ΔTHI = THI begin baseline–THI final visit; Distress trend = slope of tinnitus distress during intervention phase; ΔDistress = Baseline phase–end of intervention phase. $N = 14$.

tinnitus distress and loudness and retrospectively measured THI at the begin of baseline phase and final visit after intervention phase.

A good compliance using the app was observed, particularly regarding the diary (about 79% daily usage) and structured counseling (72% daily usage). 14% (3/21) dropped out. This high adherence was unexpected since app-based interventions in chronic diseases are normally characterized by high attrition and dropout rates [54]. This is especially true for the EMA module, because in a study with its predecessor (TrackYourTinnitus), only about 68% had more than one diary entry [55]. The authors assumed that incentives could improve adherence rates noticeably. In fact, this could have been supported empirically in other trials [35,56]. Therefore, in this study, although not intended as incentives, the intervention modules may have contributed to good user engagement, because of potential symptom improvement occurring through the usage. The finding of a more favorable compliance could have also been driven by the chosen study design with a small sample size that enabled a frequent mail contact (regarding study procedure, concerns and questions) and possibly increased the involvement of the participants. Moreover, the significance of the EMA data for research was re-emphasized before and during the study.

The THI changed statistically and clinically significant from baseline to final visit with a large effect size (Cohens $d$ = 1.1). Meta-analysis of CBT applied to tinnitus report effect sizes between 0.54 and 0.91 [14,57]. However, caution should be taken in the interpretation of this effect. First the THI change occurred during the baseline and the intervention phase and can thus not be solely attributed to the intervention. Second, data come from a pilot study with a small sample size and no control group. Larger, controlled samples are needed to validate the effect on THI. Regarding the individual level, 13 of 18 (72%) improved clinically meaningful ($\Delta$THI $\geq$ 7) [51]. This reflects the well-known heterogeneity of treatment response in tinnitus patients, i.e. tinnitus distress decreases only for some individuals, but not for the whole sample. This could also be observed in visualization of individual symptom courses and in comparisons of baseline phase with end of intervention phase. Future research should aim to identify predictors of response to treatment, especially for app-based interventions [58]. Previous evidence showed that the positive correlation between working alliance and treatment outcome known from face-to-face psychological therapy remains stable for internet-based treatments [59]. Moreover, the acceptability of internet-based interventions is a considerable predictor of treatment outcomes [60,61]. Hence, these factors should be considered in subsequent studies testing app-based tinnitus treatments.

The main analysis focus was on the EMA data (tinnitus distress and loudness). The heterogeneity between the cases and the fluctuation of symptom courses hampered identification of symptom improvement from visualization. High fluctuation has already been observed in prior EMA studies measuring tinnitus symptoms using mobile devices and reflects their usual time course [23,24,62]. Fluctuations in tinnitus symptoms are probably influenced by a combination of many factors, including neurobiological markers, circadian rhythm, behavioral and emotional processes, and stress [63–67].

To overcome the obstacle of heterogeneity and symptom fluctuation, individual means and standard deviations were calculated for the baseline phase and the end of the intervention phase (same time window than baseline phase). On a group level, neither tinnitus distress nor loudness showed a significant change from baseline to the end of intervention phase. On an individual level, however, 5 of 14 cases improved clinically meaningful in tinnitus distress ($\Delta$Distress $\geq$ 10) as well as in the THI score ($\Delta$THI $\geq$ 7). Thus, 36% of the sample experienced a reduction in tinnitus burden during the intervention. Interestingly, more patients improved clinically meaningful in the THI than in EMA of tinnitus distress (72% vs. 36%). This might indicate that the THI is more sensitive to change than EMA of tinnitus distress. Regarding

tinnitus loudness, only 2 of 14 cases (14%) improved clinically meaningful from baseline to the end of intervention phase ($\Delta$Loudness $\geq$ 10).

As visual inspection of the EMA data suggested a dissociation of tinnitus distress and loudness scores over the course of the study (see Fig 3), correlations scores between distress and loudness were calculated for the baseline phase and for each week of the intervention phase. Fig 4 depicts a descriptive decline in the correlation between tinnitus distress and loudness over the study period. This could reflect a psychological process that was triggered by the information about tinnitus received from structured counseling and the provided tips that helped the patients to better cope with their tinnitus. In other words, the patients have learned to evaluate their tinnitus in a more differentiated way, i.e. more independent of acoustic features and as a result, tinnitus distress became more independent of tinnitus loudness. Prior studies identified tinnitus acceptance and emotional valence as partial mediators of the relationship between tinnitus distress and loudness [64,68]. If the app-based intervention induces a change in those mediators could be of interest for subsequent studies.

To predict daily measured tinnitus distress, a mixed effect model was set up including demographics (age, gender), psychological burden (baseline THI and PHQ-9), intervention effects (trend, level) as fixed effects and participants as random intercepts. The model could explain 62% of the variance of the daily tinnitus distress and was checked for overfitting. Cases with higher daily distress tended to be male with higher baseline THI and PHQ-9 scores. Whereas the influence of THI and PHQ-9 on predicting the tinnitus distress score is not surprising, the gender effect was somehow unexpected, as female patients tend to report higher levels of distress than male in other studies [69,70]. This discrepancy could be due to our small sample size which may not be representative for the tinnitus population. Regarding the treatment effect, tinnitus distress is significantly reduced by 0.06 points per day (trend effect). Thus, considering a study length of about 100 days (84 days intervention phase, 8–19 days baseline phase), tinnitus distress is reduced by 6 units in total (scale: 0–100). This effect is rather weak when compared with the THI which was reduced by almost 12 units (scale: 0–100). The level effect did not reach statistical significance.

Importantly, both the trend effect in tinnitus distress and the THI effect consider the whole study period, i.e. baseline and intervention phase. Thus, presumably in some patients, tinnitus distress improved already to some extent during the baseline phase. This may indeed explain why no group effect was found for tinnitus distress (EMA), as the whole baseline phase was summarized as a contrast to the end of the intervention phase. An improvement during baseline phase could describe an anticipation effect, that is participants experience symptom improvement before the intervention has started because they expect relief from the upcoming intervention [71]. Another possibility is that the daily reflection of one's own emotional state through EMA somehow contributed to the improvement in tinnitus distress, although prior findings didn't assume any influence [23,24,35].

Finally, we compared the improvement in THI (pre-post clinical questionnaire) with improvement in tinnitus distress (EMA). $\Delta$THI was strongly associated with the Distress trend ($r$ = -0.75, $p$ < 0.01) and $\Delta$Distress ($r$ = 0.86, $p$ < 0.001). Despite high correlations, previous findings suggested that the THI could be more sensitive to change than EMA of tinnitus distress. Different thresholds for retrospective questionnaires and EMA might be appropriate. Even though, further validation research is needed to examine whether EMA is feasible to detect relevant changes in tinnitus symptoms and to validate clinical trials. In depression research, EMA has already been compared to standard assessment instruments [72,73]. In those studies, it was shown that EMA did not perform worse than paper-pencil questionnaires using the same items with respect to sensitivity to change. The authors even argued that assessing symptoms with EMA methodology may be more accurate than pre-post questionnaires

that reflect just a random time point overlooking the complex clinical picture. The high symptom fluctuation observed here underlines this point. Note that for the Distress trend only the intervention phase was considered, because we did not expect any relevant symptom changes during baseline phase. Thus, an improvement during baseline phase could explain the slightly weaker correlation of the Distress trend with ΔTHI than ΔDistress revealed.

The results in this study should be interpreted alongside considerations of its limitations. First of all, it is a pilot study with a small sample size testing the app-based intervention for the first time, thus, further studies with larger samples sizes are required to confirm its efficacy. Furthermore, due to the online and app-based study environment, data assessment and app usage could not be controlled with respect to correct app usage, user identity, and circumstances of EMA recordings. The latter should be paid special attention, because the given situation might influence the perception of tinnitus, at least its loudness. Considering future research, an implemented sensor detecting the ambient sound level might reduce this bias. Although the natural study setting leads to reduced internal validity, it increases ecological validity on the other hand. Moreover, there was no THI assessment after the baseline phase before the intervention phase started to distinguish effects between the phases. Furthermore, we have no data on long-term outcomes nor from severely affected patients. Nonetheless, based on available evidence one could speculate, that long-term outcomes and the inclusion of more severely affected patients would have increased the observed effect: counseling takes some time to evolve its whole effect and higher baseline scores in tinnitus questionnaires predict better treatment outcomes [27,29,33]. Regarding the EMA subset, we had to exclude 4 more patients because of data not missing at random. This should be considered when interpreting and comparing group effects on THI and tinnitus distress, although the data sets are comparable regarding participant demographics and tinnitus characteristics.

In conclusion, we found good treatment compliance, a meaningful improvement in tinnitus distress and tinnitus handicap for part of the sample and a weakening of the relationship between tinnitus distress and loudness over the course of the intervention. Also, there is a strong correlation between improvement in THI and improvement in EMA of tinnitus distress, however, sensitivity to change could differ and separate thresholds might be appropriate. Importantly, these results should be interpreted alongside the fact that the treatment is unguided, resource-efficient, location-independent and immediately available. On a large scale, it could provide relief for a considerable number of tinnitus patients without any additional cost or effort.

## Supporting information

**S1 Checklist. CONSORT checklist.**
(DOCX)

**S1 Protocol. Study protocol.**
(DOCX)

**S1 Table. EMA questions.**
(DOCX)

**S2 Table. EMA subset demographics.**
(DOCX)

## Acknowledgments

We want to thank Susanne Staudinger and Jan Musman who helped implement and conduct the study.

## Author Contributions

**Conceptualization:** Milena Engelke, Winfried Schlee.

**Data curation:** Carsten Vogel, Rüdiger Pryss.

**Formal analysis:** Milena Engelke, Jorge Simões.

**Funding acquisition:** Stefan Schoisswohl, Winfried Schlee.

**Investigation:** Milena Engelke.

**Project administration:** Milena Engelke, Stefan Schoisswohl.

**Software:** Carsten Vogel, Rüdiger Pryss.

**Supervision:** Jorge Simões, Thomas Probst, Berthold Langguth, Winfried Schlee.

**Visualization:** Milena Engelke.

**Writing – original draft:** Milena Engelke.

**Writing – review & editing:** Jorge Simões, Carsten Vogel, Stefan Schoisswohl, Martin Schecklmann, Stella Wölflick, Rüdiger Pryss, Thomas Probst, Berthold Langguth, Winfried Schlee.

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
