## [Decision Letter · Decision Letter 0]

8 Nov 2022

PDIG-D-22-00233

Pilot study of a smartphone-based tinnitus therapy using structured counseling and sound therapy: A multiple-baseline design with ecological momentary assessment

PLOS Digital Health

Dear Dr. Engelke,

Thank you for submitting your manuscript to PLOS Digital Health. We invite you to submit a revised version of the manuscript that addresses the points raised during the review process.

Please submit your revised manuscript within 30 days Dec 08 2022 11:59PM. If you will need more time than this to complete your revisions, please reply to this message or contact the journal office at digitalhealth@plos.org. Please include the following items when submitting your revised manuscript:

We look forward to receiving your revised manuscript.

Kind regards,

Padmanesan Narasimhan, MBBS MPH PhD

Section Editor

PLOS Digital Health

Journal Requirements:

a. Please clarify all sources of funding (financial or material support) for your study. List the grants (with grant number) or organizations (with url) that supported your study, including funding received from your institution. 

b. State the initials, alongside each funding source, of each author to receive each grant.

c. State what role the funders took in the study. If the funders had no role in your study, please state: “The funders had no role in study design, data collection and analysis, decision to publish, or preparation of the manuscript.”

d. If any authors received a salary from any of your funders, please state which authors and which funders.

2. We ask that a manuscript source file is provided at Revision. Please upload your manuscript file as a .doc, .docx, .rtf or .tex.

4. In the online submission form, you indicated that your data will be submitted to a repository upon acceptance. We strongly recommend all authors deposit their data before acceptance, as the process can be lengthy and hold up publication timelines. Please note that, though access restrictions are acceptable now, your entire data will need to be made freely accessible if your manuscript is accepted for publication. This policy applies to all data except where public deposition would breach compliance with the protocol approved by your research ethics board. If you are unable to adhere to our open data policy, please kindly revise your statement to explain your reasoning and we will seek the editor's input on an exemption. Please be assured that, once you have provided your new statement, the assessment of your exemption will not hold up the peer review process.

Additional Editor Comments (if provided):

Dear Authors

We have received comments from the reviewers. Can you please address them to proceed further?

Reviewers' comments:

Reviewer's Responses to Questions

**Comments to the Author**

1. Does this manuscript meet PLOS Digital Health’s publication criteria? Is the manuscript technically sound, and do the data support the conclusions? The manuscript must describe methodologically and ethically rigorous research with conclusions that are appropriately drawn based on the data presented.

Reviewer #1: Yes

Reviewer #2: Yes

Reviewer #3: Yes

2. Has the statistical analysis been performed appropriately and rigorously?

Reviewer #1: Yes

Reviewer #2: N/A

Reviewer #3: Yes

3. Have the authors made all data underlying the findings in their manuscript fully available (please refer to the Data Availability Statement at the start of the manuscript PDF file)?

Reviewer #1: Yes

Reviewer #2: No

Reviewer #3: Yes

4. Is the manuscript presented in an intelligible fashion and written in standard English?

Reviewer #1: Yes

Reviewer #2: Yes

Reviewer #3: Yes

5. Review Comments to the Author

Reviewer #1: This rigourous pilot study evaluated the efficacy of a smartphone application consisting of educational counseling and sound therapy for tinnitus patients.

I only suggest to improve the inclusion/exclusion criteria of participants and to add some details in the methods.

Did you include patients with normal hearing or hearing loss? Did you include patients with pulsatile tinnitus, somatosensory tinnitus? 

As you mentioned 7 patients didn't received a clinical evaluation, could you add clinical data for the other 11 patients (normal hearing, Meniere's disease, depression/anxiety, ecc.)?

How many patients with hyperacusis based on GUEF?

Did you evaluate the participants during the COVID-19 pandemic?

Reviewer #2: The study entitled “Pilot study of a smartphone-based tinnitus therapy using structured counseling and sound therapy: A multiple-baseline design with ecological momentary assessment” is concerned on the feasibility study to evaluate an app-based intervention. A small sample of, originally 20 people, underwent the intervention. The well-known Tinnitus handicap Inventory (THI) was used before and after the intervention together with a system to collect ecological momentary assessments (EMA) in a daily basis about the self-perceived tinnitus loudness and distress. The results showed: 1) A very good treatment compliance and the use of the app; 2) An improvement of the THI score after the intervention; 3) A trend of improvement in the EMA evaluations. 4) A correlation between the difference in THI and Distress from the EMA. The interpretations of the results, perspectives and limitations are adequate for this study. The authors discuss the results being aware of the small sample size and being cautious on their interpretations. 

Major points

The manuscript is well written, the motivation, methods and rationale are clear and well explained. The analyses are appropriate and include some practical aspects. The interpretations and discussions are cautious and interesting for such a pilot study. However, I have some questions to be addressed by the authors.

1) The multiple-baseline design is unclear to me. I do not understand what the motivation behind this stratification is. What are the hypotheses? Or, is there a future use of this beyond this pilot? L128-135 explain the use of the multiple-baseline design. However, the results do not make use of this stratification and it is not discussed either.

2) The fundamentals of Ecological momentary assessments involve not only the use of repeated assessments. It is of great importance to gather information about the situation, that is why the word “ecological” is included. In the present study I can see two flaws on that respect. 1) There is no information gathered about the situation where the patients are when they answer the EMA. Are they at home in a quiet place? Are they in public transport? The situation might very much affect the tinnitus perception, at least in terms of loudness. 2) There is a reminder every day at the same time and this might have affect the “moment” when the patients made use of the app, both the EMA and the intervention. Also, it seems that it was limited to only one EMA per day. As it is right now, it looks more like a “daily Momentary assessment” than a formal EMA. The authors can discuss this, the decisions taken and the implications of those decisions. 

Specific comments

L27: No need of the trial registration in the abstract. 

L33-34: Perhaps too specific to use the Cohen’s effect size in the abstract.

L37: Do you mean that at the end of the study the loudness might be the same but the distress is lower? (So that is why the correlation is lower).

L39: I think it should be r= 0.86 because it is the change in EMA 

L50: we evaluated

L52: compliance and efficacy. This looks important and could be also part of the abstract.

L53: baseline and “at the” end

L58-59: Already said in the abstract.

L55-58: Since the sample is small, consider using (X number of participants)/(N total of participants) instead of percentages. This applies to the entire text.

L91-93: I don’t think the mention of COVID19 is relevant here. In general terms, the authors could mention that environmental factors might limit the access to face to face treatments as it happened, for example, in 2020 due to COVID 19. 

L98: In response to “that”. What is “that”?

L98: What do you mean by low threshold? Please, explain.

L100-101. They are or they could be particularly beneficial for tinnitus? Is there already evidence in the case of tinnitus? I suggest removing “such as tinnitus”. Also the references do not mention tinnitus.

L110: Is there any evaluation of CBT Apps in other field that you could refer to?

L138: Congruent? What does that mean?

L142: Staggered introduction seems to be the same as multiple baseline design.

L213-215: Caption of Fig. 1: Why do you use “It should… It should…, It should have been daily used”. Confusing wording.

L240: THI already defined before. 

L249: Is it possible to include these specific questions in an appendix?

L253: Did the daily answers occurred between 19:30-20:00 in the majority of the cases? In that case the EMA might have been influenced by having a reminder every day at the same time.

L264-265: Did you show these results? How did you compare the groups, in different pairs? Perhaps, I got confused and you mean the results in Table 2. 

L269-271: I think it is unnecessary to explain what was plotted.

L287: (14%). It is preferable to write 3/21 or 3 out of 21.

Fig 3.: Since the sample is small and the results might not follow a normal distribution. I suggest you use boxplots instead. Besides, you can use different symbols to show the people in each group. Also, the Xlabel can be “Moment of the intervention” for example.

L333: So, only 14 out of 18 (77%) or 14 out of 21 (66%). 

L362-364 and Table 3 and 4: All the \\DeltaDistress, \\DeltaTHI etc. When the number is positive, does it mean that the score in the Final visit is larger than in the baseline? Usually, when using ‘change’ (\\Delta), it should be End of Intervention phase – Baseline phase. Looking at the footnote of Table 3 I see that here you use a different convention. Therefore, you might want to say that \\DeltaXXX is improvement rather than change or just change the signs for consistency. Also, it would be good to remind the readers that a higher number means more handicap, distress and loudness.

L407-409. The authors use a mixed linear model as if it was a regression model, to my understanding, these are mathematically similar, but they have different applications. While the first one is usually fitted to the entire dataset and used for exploring effects of factors, regression is used for predictions, so it is important to avoid overfitting and to test the model. Since this is a pilot study, I believe the first approach would be sufficient to explore the effects of the different factors. Alternatively, iterative resampling can be used, and the ranges of the coefficients and p-values used for the analysis. 

L434- With an initial sample of 21 ….

L442 – Were there incentives in this study?

L445 – Please elaborate this. Do you mean the user experience? Is it because the patients can get a benefit from the intervention?

L446 – the frequent mail contact was only the daily reminder or also some communication. 

L449-454- Good and cautious interpretations.

L455 – 72% (X/N?)

L470 – What about Simoes, J., Bulla, J., Neff, P., Pryss, R., Marcrum, S. C., Langguth, B., & Schlee, W. (2022). Daily Contributors of Tinnitus Loudness and Distress: An Ecological Momentary Assessment Study. Frontiers in Neuroscience, 16, 883665. https://doi.org/10.3389/fnins.2022.883665 ?

L473 – What is the obstacle?

L480 – (72% vs 36%). Is it worth discussing that THI corresponds to several questions about participation restrictions and emotional distress and the EMA distress is just one question?

L491-496: Unclear paragraph

L522 - change in tinnitus distress from EMA.

L522-523-The values of the correlations might be unnecessary.

L532 – Consider referencing: Andersson, K. E., Andersen, L. S., Christensen, J. H., & Neher, T. (2021). Assessing Real-Life Benefit From Hearing-Aid Noise Management: SSQ12 Questionnaire Versus Ecological Momentary Assessment With Acoustic Data-Logging. American Journal of Audiology, 30(1), 93–104. https://doi.org/10.1044/2020_AJA-20-00042

L538-541. Repetitive. Perhaps move it to the concluding paragraph.

L543.- By circumstances… do you mean the situations?

L560 -561- Unclear.

Reviewer #3: Authors conducted a pilot study to investigate the feasibility and efficacy of an app-based tinnitus intervention combining structured counseling with acoustic therapy using a multiple-baseline across groups design with a sample of 21 participants. Outcome variables were EMA of tinnitus distress and loudness and retrospectively measured THI at the begin of baseline phase. Authors indicate that app-based structured counseling combined with sound therapy is feasible, has an impact on tinnitus symptoms and reduces distress for several patients. In addition, they suggest that EMA could be used as a measurement tool to detect changes in tinnitus symptoms in clinical trials as has already been shown in other areas of mental health research. Manuscript is well-written and reasonable, valuable research. There were minor concerns to publish.

Minor concerns

1. Introduction is too long to read. Please be short.

2. I cannot understand the meanings of 4 groups. Please explain easily by Figure.

6. PLOS authors have the option to publish the peer review history of their article (what does this mean?). If published, this will include your full peer review and any attached files.

**Do you want your identity to be public for this peer review?** For information about this choice, including consent withdrawal, please see our Privacy Policy.

Reviewer #1: Yes: Alessandra Fioretti

Reviewer #2: Yes: Raul Sanchez-Lopez

Reviewer #3: No

---

## [Decision Letter · Decision Letter 1]

13 Dec 2022

Pilot study of a smartphone-based tinnitus therapy using structured counseling and sound therapy: A multiple-baseline design with ecological momentary assessment

PDIG-D-22-00233R1

Dear Ms. Engelke,

We are pleased to inform you that your manuscript 'Pilot study of a smartphone-based tinnitus therapy using structured counseling and sound therapy: A multiple-baseline design with ecological momentary assessment' has been provisionally accepted for publication in PLOS Digital Health.

Best regards,

Padmanesan Narasimhan, MBBS MPH PhD

Section Editor

PLOS Digital Health

Reviewer Comments (if any, and for reference):

Reviewer's Responses to Questions

**Comments to the Author**

1. If the authors have adequately addressed your comments raised in a previous round of review and you feel that this manuscript is now acceptable for publication, you may indicate that here to bypass the “Comments to the Author” section, enter your conflict of interest statement in the “Confidential to Editor” section, and submit your "Accept" recommendation.

Reviewer #1: All comments have been addressed

Reviewer #2: All comments have been addressed

Reviewer #3: All comments have been addressed

2. Does this manuscript meet PLOS Digital Health’s publication criteria? Is the manuscript technically sound, and do the data support the conclusions? The manuscript must describe methodologically and ethically rigorous research with conclusions that are appropriately drawn based on the data presented.

Reviewer #1: Yes

Reviewer #2: Yes

Reviewer #3: Yes

3. Has the statistical analysis been performed appropriately and rigorously?

Reviewer #1: Yes

Reviewer #2: Yes

Reviewer #3: Yes

4. Have the authors made all data underlying the findings in their manuscript fully available (please refer to the Data Availability Statement at the start of the manuscript PDF file)?

Reviewer #1: Yes

Reviewer #2: Yes

Reviewer #3: Yes

5. Is the manuscript presented in an intelligible fashion and written in standard English?

Reviewer #1: Yes

Reviewer #2: Yes

Reviewer #3: Yes

6. Review Comments to the Author

Reviewer #1: (No Response)

Reviewer #2: Dear Authors;

Thank you for taking my comments into consideration for the revision of the manuscript. After going through the response, one by one, and read again the manuscript, I think this is a relevant work for the readers of PLOS Digital Health. I must say that I expected a more thorough discussion about the fundamentals of Ecologically momentary assessments that I pointed out in the previous review (i.e., that it corresponds to specific time and environment), which are not fully reflected in your study design. However, the study is sound, it fulfils the criteria of PLOS Digital Health and this comment will be part of the peer-review reports even if the authors did not discuss it in the main text.

Reviewer #3: All comments were corrected.

7. PLOS authors have the option to publish the peer review history of their article (what does this mean?). If published, this will include your full peer review and any attached files.

**Do you want your identity to be public for this peer review?** For information about this choice, including consent withdrawal, please see our Privacy Policy.

Reviewer #1: **Yes: **Alessandra Fioretti

Reviewer #2: **Yes: **Raul Sanchez-Lopez

Reviewer #3: **Yes: **Toru Miwa
